# Risk factor analysis and creation of an externally-validated prediction model for perioperative stroke following non-cardiac surgery: A multi-center retrospective and modeling study

Yulong Ma[1,2,ᵒ], Siyuan Liu[1,2,3,ᵒ], Faqiang Zhang[1,ᵒ], Xuhui Cong[4,ᵒ], Bingcheng Zhao[5,ᵒ], Miao Sun[1], Huikai Yang[1], Min Liu[6], Peng Li[7], Yuxiang Song[1], Jiangbei Cao[1], Yingfu Li[1], Wei Zhang[4], Kexuan Liu[5]*, Jiaqiang Zhang[4]*, Weidong Mi[1,2]*

1 Department of Anesthesiology, The First Medical Center of Chinese PLA General Hospital, Beijing, China, 2 Nation Clinical Research Center for Geriatric Diseases, Chinese PLA General Hospital, Beijing, China, 3 Department of Anesthesiology, Affiliated Hospital of Nantong University, Nantong, China, 4 Department of Anesthesia and Perioperative Medicine, Henan Provincial People's Hospital and People's Hospital of Zhengzhou University, Zhengzhou, China, 5 Department of Anesthesiology, Nanfang Hospital, Southern Medical University, Guangzhou, China, 6 Department of Anesthesiology, Beijing Tongren Hospital, Capital Medical University, Beijing, China, 7 Department of Anesthesiology, The Sixth Medical Center of Chinese PLA General Hospital, Beijing, China

ᵒ These authors contributed equally to this work.
* liukexuan705@163.com (KL); jqzhang@henu.edu.cn (JZ); wwdd1962@163.com (WM)

## Abstract

### Background

Perioperative stroke is a serious and potentially fatal complication following non-cardiac surgery. Thus, it is important to identify the risk factors and develop an effective prognostic model to predict the incidence of perioperative stroke following non-cardiac surgery.

### Methods and findings

We identified potential risk factors and built a model to predict the incidence of perioperative stroke using logistic regression derived from hospital registry data of adult patients that underwent non-cardiac surgery from 2008 to 2019 at The First Medical Center of Chinese PLA General Hospital. Our model was then validated using the records of two additional hospitals to demonstrate its clinical applicability. In our hospital cohorts, 223,415 patients undergoing non-cardiac surgery were included in this study with 525 (0.23%) patients experiencing a perioperative stroke. Thirty-three indicators including several intraoperative variables had been identified as potential risk factors. After multi-variate analysis and stepwise elimination ($P < 0.05$), 13 variables including age, American Society of Anesthesiologists (ASA) classification, hypertension, previous stroke, valvular heart disease, preoperative steroid hormones, preoperative β-blockers, preoperative mean arterial pressure, preoperative fibrinogen to albumin ratio, preoperative fasting plasma glucose, emergency surgery, surgery type and surgery length were screened as independent risk

**Data availability statement:** The data used and analyzed during the current study are not freely available for the ethical restriction, because the data contain potentially identifying and sensitive patient information. But, the data are available from Department of Medical Service, The First Medical Center of Chinese PLA General Hospital upon reasonable request (Email: 15776734388@163.com and Tel: 086 010 66938152).

**Funding:** W.M. was supported by the National Key Research and Development Program of China (no. 2018YFC2001901), and the National Natural Science Foundation of China (No. 81801193). Y.M. was supported by the Capital Health Research and Development of Special (2022-4-5025), and the National Natural Science Foundation of China (No. 82171464; No. 82371469). The funders had no role in study design, data collection and analysis, decision to publish, or preparation of the manuscript.

**Competing interests:** The authors have declared that no competing interests exist.

**Abbreviations:** ACEIs, angiotensin-converting enzyme inhibitors; ARB, angiotensin II receptor blockers; ASA, American Society of Anesthesiologists; AUC, area under the receiver operating characteristic curve; BMI, body mass index; CI, confidence interval; DCA, Decision curve analysis; FAR, fibrinogen to albumin ratio; FPG, fasting plasma glucose; ICD9, International Classification of Diseases 9; MAP, mean arterial pressure; NBs, net benefits; NLR, neutrophil-lymphocyte ratio; PLR, platelet-to-lymphocyte ratio; TRIPOD, Transparent Reporting of a multivariable prediction model for Individual Prognosis or Diagnosis; VIFs, variance inflation factors; 301PSRC, 301 Perioperative Stroke Risk Calculator.

factors and incorporated to construct the final prediction model. Areas under the curve were 0.893 (95% confidence interval (CI) [0.879, 0.908]; $P < 0.001$) and 0.878 (95% CI [0.848, 0.909]; $P < 0.001$) in the development and internal validation cohorts. In the external validation cohorts derived from two other independent hospitals, the areas under the curve were 0.897 and 0.895. In addition, our model outperformed currently available prediction tools in discriminative power and positive net benefits. To increase the accessibility of our predictive model to doctors and patients evaluating perioperative stroke, we published an online prognostic software platform, 301 Perioperative Stroke Risk Calculator (301PSRC). The main limitations of this study included that we excluded surgical patients with an operation duration of less than one hour and that the construction and external validation of our model were from three independent retrospective databases without validation from prospective databases and non-Chinese databases.

## Conclusions

In this work, we identified 13 independent risk factors for perioperative stroke and constructed an effective prediction model with well-supported external validation in Chinese patients undergoing non-cardiac surgery. The model may provide potential intervention targets and help to screen high-risk patients for perioperative stroke prevention.

## Author summary

### Why was this study done?

- Perioperative stroke can have devastating consequences. It is crucial to enhance perioperative management for high-risk patients to minimize the occurrence of perioperative stroke and improve clinical outcomes.

- Preventative care for at-risk patients for perioperative stroke requires the identification of underlying risk factors.

- A specific predictive model could accurately screen for high-risk patients before surgery in order to inform perioperative clinical interventions to reduce the incidence of perioperative stroke.

### What did the researchers do and find?

- We identified thirteen independent risk factors for perioperative stroke which include patient age, American Society of Anesthesiologists (ASA) classification, hypertension, previous stroke, valvular heart disease, preoperative steroid hormones, preoperative β-blockers, preoperative mean arterial pressure (MAP), preoperative fibrinogen to albumin ratio (FAR), preoperative fasting plasma glucose (FPG), emergency surgery, surgery type and surgery length.

- We developed and internally validated a specific and accurate model to predict the incidence of perioperative stroke from 223,415 Chinese patients undergoing non-cardiac surgery, and externally well validated the model in another two Chinese medical centers.

- We developed an evaluation software named 301 Perioperative Stroke Risk Calculator (301PSRC) to screen for high-risk patients before surgery.

**What do these findings mean?**

- Our findings provide several potential intervention targets (preoperative MAP, preoperative FAR, preoperative FPG, and medication optimization) for perioperative stroke prevention.

- Our findings indicate that the prediction model has promising potential to improve prediction of perioperative stroke.

- Limitations were that we excluded surgical patients with an operation duration of less than one hour, and that the construction and external validation of our model were from three independent retrospective databases without validation from prospective databases and non-Chinese databases.

## Introduction

Perioperative stroke refers to any cerebrovascular event characterized by motor, sensory, or cognitive dysfunction caused by embolism, thrombosis, or bleeding, occurring either during surgery or within 30 days post-surgery [1]. As such, this complication can have devastating consequences for the patient. Similar to non-perioperative strokes, most perioperative strokes are ischemic rather than hemorrhagic [2]. In patients experiencing perioperative stroke following non-cardiac surgery, the rate of disability or death can be as high as 84%, which is mainly due to delayed treatment, diagnosis, infrequent intervention, and limited use of thrombolysis to address potential blockage following recent surgery [3]. Consequently, it is crucial to enhance perioperative management for high-risk patients to minimize the occurrence of perioperative stroke and improve clinical outcomes.

Preventative care for at-risk patients for perioperative stroke requires the identification of underlying risk factors. Several preoperative indicators have been identified as the independent risk factors for perioperative stroke [4–8]. However, systematic analysis of the risk factors for perioperative stroke in Chinese surgical patients has not been reported before. Several instruments have been developed to identify high-risk patients including the CHA2DS2VASc score, ACS-NSQIP Surgical Risk Calculator, and the MICA Risk Score [9–12]. However, these risk estimators predicted the risk of various perioperative complications of the cerebrovascular system, which limits their predictive power for perioperative stroke in particular. Recently, a new risk model has been developed to predict postoperative stroke using data from large administrative database with relatively high predictive power [10]. However, in a recent authoritative systemic review, the author cautioned the use of this prediction model because it has not been validated externally [13]. Therefore, there is a need to develop a specific predictive model that could accurately screen for high-risk patients before surgery in order to inform perioperative clinical interventions to reduce the incidence of perioperative stroke.

This study aimed to identify new risk factors for perioperative stroke and to develop and externally validate a specific and accurate instrument to predict perioperative stroke by using a large cohort of Chinese patients undergoing non-cardiac surgery from three medical centers. Furthermore, we compared the predictive efficacy of our model with that of existing perioperative stroke risk stratification scores. According to the prediction model, we developed an evaluation software named 301 Perioperative Stroke Risk Calculator (301PSRC) for evaluating perioperative stroke.

## Methods

This was a multi-center retrospective study that was approved by the Medical Ethics Committee of The First Medical Center of Chinese PLA General Hospital (reference

number: S2019-311-03), and the requirement for informed content was exempt. External validation data were legally exempted from research ethics board review as it were already anonymized and de-identified. This study is reported in line with the Transparent Reporting of a multi-variable prediction model for Individual Prognosis or Diagnosis (TRIPOD): The TRIPOD statement (S1 Table).

## Study design and cohorts

For the creation of a predictive model, we retrieved data from all patients undergoing non-cardiac surgery between January 2008 and August 2019 at The First Medical Center of Chinese PLA General Hospital ($n$ = 376,933 total qualified patients). For external validation, we used data of non-cardiac surgical inpatients from Nanfang Hospital (between January 2019 and October 2021, $n$ = 62,407 total qualified patients) and Henan Provincial People's Hospital (between December 2014 and June 2021, $n$ = 412,043 total qualified patients). The current study included all the non-cardiac surgeries during the research period, such as ear, nose and throat, obstetrics and gynecology, abdominal surgery, orthopedics, stomatology, urology, general surgery, neurosurgery, thoracic surgery, vascular surgery, and other surgeries (trauma surgery and plastic surgery). To avoid repeated data use, for patients with multiple surgeries during one hospitalization period, only the first surgery was included in our analysis. Exclusion criteria were patients under the age of 18 years old, with surgery duration of less than one hour, those who underwent regional anesthesia, American Society of Anesthesiologists (ASA) classification ≥ V, or high levels (55% or above) of missing variables. The external cohorts followed the same exclusion criteria as the development cohort.

## Outcome

The outcome of interest was a perioperative ischemic stroke, which was defined as a new-onset brain infarction during their hospital stay within 30 days following surgery (excluding the day of surgery). Neuroimaging was performed on patients suspected with cerebral stroke when typical symptoms and signs appeared after surgery. Stroke diagnoses were confirmed by a combination of neuroimaging and clinical evidence of cerebrovascular ischemia as identified through International Classification of Diseases 9 (ICD9)/ICD10 diagnosis codes (S2 Table). The outcome variable was validated through a medical record review by the neurologists.

## Predictor variables

We specified candidate predictor variables based on published studies and established clinical knowledge including baseline characteristics, comorbidities, pre-operative medication, pre-operative laboratory tests, and surgery-related variables. Pre-operative laboratory tests were performed by the central laboratory. Preoperative mean arterial pressure (MAP) was calculated based on the pressure value measured at the time of preoperative evaluation. Variables with ≥20% missing data were excluded. We used multiple imputations to impute missing values for variables with <20% missing data by the classification and regression tree (cart) method.

## Model development and validation

The prediction model was constructed based on the data from The First Medical Center of Chinese PLA General Hospital. Patients were randomly divided into a training dataset and an internal validation dataset with a split ratio of 70% and 30% of total patients. The training dataset was used to develop the prediction model in the final logistic regression, whereas the

validation dataset was used for internal validation. A univariate analysis was performed, and candidate variables with a *P*-value less than 0.05 were included in the multi-variable model.

Dependent variables including age, ASA classification, body mass index (BMI), hypertension, diabetes mellitus, myocardial infarction, coronary heart disease, heart failure, atrial fibrillation, previous stroke, valvular heart disease, angina pectoris, peripheral vascular disease, malignant tumor, preoperative serum albumin, preoperative fasting plasma glucose (FPG), preoperative angiotensin-converting enzyme inhibitors (ACEIs), preoperative angiotensin II receptor blockers (ARB), preoperative steroids, preoperative β-blockers, preoperative calcium channel blockers, preoperative MAP, emergency surgery, surgery type, surgery length, preoperative neutrophil-lymphocyte ratio (NLR), preoperative platelet-to-lymphocyte ratio (PLR), preoperative fibrinogen to albumin ratio (FAR), blood product usage, crystals volume, and opioids dosage were collected for the multi-variate analysis.

Stepwise elimination was performed with $P < 0.05$ for retaining a given predictor in the model. But, for preoperative β-blockers ($P = 0.093$) and preoperative steroids ($P = 0.057$), we included the two variables in the final model, considering the clinical significance. We also assessed potential multi-collinearity amongst all predictors in the final model using variance inflation factors (VIFs), and VIF less than 4 was considered to represent no significant collinearity. Model discrimination was measured using the area under the receiver operating characteristic curve (AUC). Model calibration was assessed using the Hosmer–Lemeshow goodness of fit test. Decision curve analysis (DCA) was used to demonstrate the net benefits (NBs) with each threshold probability.

External validation was performed using patients' data from Nanfang Hospital and Henan Provincial People's Hospital. The training dataset was also used to validate the existing prognostic model proposed by Mashour and colleagues [5], the ATRIA stroke risk score [14], the model proposed by Lip and colleagues [15], the model proposed by Wolf and colleagues [16], and the model proposed by Woo and colleagues [10]. The prediction model was only compared with the ATRIA stroke risk score and the model proposed by Lip and colleagues using the data of Nanfang Hospital and Henan Provincial People's Hospital, as the prediction variables—current smoker, chronic obstructive pulmonary disease, hematocrit, and preoperative serum sodium in the other three models were absent in the two external cohorts.

## Statistical analyses

The risk prediction model was built based on the data set from The First Medical Center of Chinese PLA General Hospital ($n = 223,415$). The data from Nanfang Hospital ($n = 37,568$) and Henan Provincial People's Hospital ($n = 48,719$) were used to validate the model. Predictor variables were compared using $\chi^2$ ($n \geq 5$) or Fisher's exact tests ($n < 5$) for categorical variables. Normally distributed continuous variables were compared using analysis of variance, and skewed continuous variables were compared using Kruskal–Wallis tests. Categorical data were reported as frequencies (percentages), and continuous variables were reported as medians (quartiles). We constructed the logistic regression model with perioperative ischemic stroke as the dependent variable. Data analyses were performed in R version 4.1.2 with R studio, along with the stringr, mice, car, rms, MASS, pROC, nomogramFormula, verification, ResourceSelection, and rmda packages.

## Results

### Study cohorts and perioperative stroke rate

We screened a total of 376,933, 62,304, and 412,043 patients at The First Medical Center of Chinese PLA General Hospital, Nanfang Hospital, and Henan Provincial People's Hospital,

which yielded 223,415, 37,568, and 48,719 eligible patients respectively, aged from 18 to 99 years old (S1 Fig). A total of 525 (0.23% of 223,415), 122 (0.32% of 37,568), and 173 (0.36% of 48,719) patients experienced perioperative ischemic stroke, respectively. Patients that developed perioperative ischemic stroke were generally older and experienced longer surgery duration in comparison to those who did not suffer perioperative ischemic stroke (64-years old versus 52-years old, $P < 0.001$; 195 min versus 148 min, $P < 0.001$, respectively). The prevalence of hypertension, diabetes, and previous ischemic stroke and peripheral vascular disease was also higher in patients experiencing perioperative stroke ($P < 0.001$ for each comparison). The baseline characteristics of the participants from The First Medical Center of Chinese PLA General Hospital are shown in Table 1. The baseline characteristics of the participants from Nanfang Hospital and Henan Provincial People's Hospital are shown in S3 Table.

## Analysis of risk factors

Univariate logistic regression analysis demonstrated that risk factors for perioperative ischemic stroke included patient age, ASA classification, BMI, hypertension, diabetes mellitus, myocardial infarction, coronary heart disease, heart failure, atrial fibrillation, previous stroke, valvular heart disease, angina pectoris, peripheral vascular disease, malignant tumor, preoperative serum albumin, preoperative FPG, preoperative ACEIs, preoperative ARB, preoperative steroids, preoperative β-blockers, preoperative calcium channel blockers, preoperative MAP, emergency surgery, surgery type, surgery length, preoperative NLR, preoperative PLR, preoperative FAR, blood product usage, crystals volume, and opioids dosage (S4 Table).

## Prediction model for perioperative stroke

A total of 13 factors were classified as independent predictors of perioperative ischemic stroke in the final model. These factors were: patient age, ASA classification, hypertension, previous stroke, valvular heart disease, preoperative steroid hormones, preoperative β-blockers, preoperative MAP, preoperative FAR, preoperative FPG, emergency surgery, surgery type, and surgery length (Table 2). As analyses were adjusted in the final model, we detailed the unadjusted analyses in S5 Table. Furthermore, the rejected variables are detailed in S6 Table. The perioperative ischemic stroke predictors were incorporated into the predictive nomogram (S2 Fig). For each variable, there is a corresponding score on the top line, and the total points could be calculated to indicate the probability of perioperative ischemic stroke (S7 Table). The best cutoff value was determined by the maximum value of Youden index, of which the sensitivity was 0.874 and the specificity was 0.544. The total points above the cutoff value were classified as high risk, while the results below the cut-off value were deemed low risk. The cut-off value of high-risk probability was 144.2 points, and the corresponding percentage chance of developing stroke was 0.2%.

## Model efficiency and internal validation

The AUC in the developmental cohort was 0.893 (95% confidence interval (CI) [0.879, 0.908]; $P < 0.001$) (Fig 1A). The model demonstrates high performance in a Hosmer–Lemeshow test between the expected and observed risks, with overestimation at the extremes ($P = 0.059$) (Fig 1B), owing to the small sample size of individuals with higher stroke probability. The clinical utility of the risk model was evaluated by DCA, which mainly focused on the NBs. NB combines the number of true positives and false positives into a single "net" number and is obtained by dividing the net true positives by the sample size. It reflects the balance between the benefit of a true positive and the harm of a false positive. In the context of this study, "treated" refers to patients receiving interventions based on the prediction model. The "treat

**Table 1. Patient characteristics in the development cohorts.**

|  | No postoperative ischemic stroke | Postoperative ischemic stroke | Total |
|---|---|---|---|
| **Characteristics** | *n* = 222,890 | *n* = 525 | *n* = 223,415 |
| **Age, years** | 52 (41, 62) | 64 (55, 71) | 52 (41, 62) |
| **Sex** |  |  |  |
| Male | 113228 (50.8) | 275 (52.4) | 113503 (50.8) |
| Female | 109662 (49.2) | 250 (47.6) | 109912 (49.2) |
| **ASA classification** |  |  |  |
| I | 32469 (14.6) | 18 (3.4) | 32487 (14.5) |
| II | 170443 (76.5) | 318 (60.6) | 170761 (76.4) |
| III | 18145 (8.1) | 148 (28.2) | 18293 (8.2) |
| IV | 1833 (0.8) | 41 (7.8) | 1874 (0.8) |
| **BMI, kg/m$^2$** | 24.2 (21.9, 26.6) | 24.61 (22.8, 26.9) | 24.2 (21.9, 26.6) |
| **Hypertension, *n* (%)** | 43940 (19.7) | 273 (52) | 44213 (19.8) |
| **Diabetes, *n* (%)** | 27492 (12.3) | 151 (28.8) | 27643 (12.4) |
| **Myocardial infarction, *n* (%)** | 902 (0.4) | 10 (1.9) | 912 (0.4) |
| **Coronary heart disease, *n* (%)** | 8119 (3.6) | 65 (12.4) | 8184 (3.7) |
| **Heart failure, *n* (%)** | 204 (0.1) | 5 (1) | 209 (0.1) |
| **Atrial fibrillation, *n* (%)** | 824 (0.4) | 13 (2.5) | 837 (0.4) |
| **Previous stroke, *n* (%)** | 5052 (2.3) | 163 (31) | 5215 (2.3) |
| **Valvular heart disease, *n* (%)** | 757 (0.3) | 7 (1.3) | 764 (0.3) |
| **Angina pectoris, *n* (%)** | 663 (0.3) | 7 (1.3) | 670 (0.3) |
| **Peripheral vascular disease, *n* (%)** | 8370 (3.8) | 118 (22.5) | 8488 (3.8) |
| **Renal insufficiency, *n* (%)** | 2135 (1) | 13 (2.5) | 2148 (1) |
| **Malignant tumor, *n* (%)** | 100981 (45.3) | 190 (36.2) | 101171 (45.3) |
| **Preoperative hemoglobin, g/L** | 134 (122, 146) | 132 (121, 145) | 134 (122, 146) |
| **Preoperative serum albumin, g/L** | 41.5 (39.1, 43.8) | 40.2 (37.3, 42.7) | 41.5 (39.1, 43.8) |
| **Preoperative total bilirubin, μmol/L** | 10.7 (8, 14.4) | 10.4 (8, 14.5) | 10.7 (8, 14.4) |
| **Preoperative glucose, mmol/L** | 4.89 (4.5, 5.5) | 5.53 (4.8, 7.4) | 4.89 (4.5, 5.5) |
| **Preoperative thrombin time's** | 16.3 (15.7, 17.1) | 16.2 (15.5, 17) | 16.3 (15.7, 17.1) |
| **Preoperative ACEI drugs, *n* (%)** | 4465 (2) | 32 (6.1) | 4497 (2) |
| **Preoperative ARB drugs, *n* (%)** | 8784 (3.9) | 54 (10.3) | 8838 (4) |
| **Preoperative steroids, *n* (%)** | 16052 (7.2) | 61 (11.6) | 16113 (7.2) |
| **Preoperative β-blockers, *n* (%)** | 8444 (3.8) | 69 (13.1) | 8513 (3.8) |
| **Preoperative calcium channel blockers, *n* (%)** | 26352 (11.8) | 208 (39.6) | 26560 (11.9) |
| **Preoperative MAP, mmHg** | 91.33 (83.3, 99.3) | 97.67 (90, 106) | 91.3 (83.3, 99.3) |
| **Perioperative nonsteroidal drugs, *n* (%)** | 139404 (62.5) | 397 (75.6) | 139801 (62.6) |
| **Emergency surgery, *n* (%)** | 6210 (2.8) | 77 (14.7) | 6287 (2.8) |
| **Surgery type, *n* (%)** |  |  |  |
| ENT | 21801 (9.8) | 37 (7) | 21838 (9.8) |
| Obstetrics and gynecology | 15406 (6.9) | 12 (2.3) | 15418 (6.9) |
| Abdominal surgery | 57557 (25.8) | 84 (16) | 57641 (25.8) |
| Orthopedics | 40760 (18.3) | 109 (20.8) | 40869 (18.3) |
| Stomatology | 9460 (4.2) | 17 (3.2) | 9477 (4.2) |
| Urology | 18596 (8.3) | 24 (4.6) | 18620 (8.3) |
| General surgery | 17004 (7.6) | 4 (0.8) | 17008 (7.6) |
| Other surgeries | 4699 (2.1) | 4 (0.8) | 4703 (2.1) |
| Neurosurgery | 20148 (9) | 200 (38.1) | 20348 (9.1) |
| Thoracic surgery | 15268 (6.9) | 17 (3.2) | 15285 (6.8) |

*(Continued)*

**Table 1.** (Continued)

| Characteristics | No postoperative ischemic stroke | Postoperative ischemic stroke | Total |
|---|---|---|---|
| | *n* = 222,890 | *n* = 525 | *n* = 223,415 |
| Vascular surgery | 2191 (1) | 17 (3.2) | 2208 (1) |
| **Surgery length, min** | 148 (100, 215) | 195 (135, 279) | 148 (100, 215) |
| **Amount of blood loss, ml** | 100 (50, 200) | 200 (50, 300) | 100 (50, 200) |
| **Intraoperative steroids, *n* (%)** | 180177 (80.8) | 443 (84.4) | 180620 (80.8) |
| **NLR** | 1.8 (1.4, 2.5) | 2.4 (1.7, 4.5) | 1.81 (1.4, 2.5) |
| **PLR** | 116.8 (91.6, 152.3) | 133.40 (99.1, 179.3) | 116.8 (91.6, 152.4) |
| **FAR** | 0.07 (0.06, 0.09) | 0.08 (0.07, 0.11) | 0.07 (0.06, 0.09) |
| **Blood product usage, *n* (%)** | 25942 (11.6) | 106 (20.2) | 26048 (11.7) |
| **Crystals, mL/kg/h** | 8.41 (6.2, 11.4) | 7.17 (5.2, 10) | 8.41 (6.2, 11.4) |
| **Colloids, mL/kg/h** | 2.63 (0, 4.2) | 2.59 (1.3, 3.9) | 2.63 (0, 4.2) |
| **Morphine equivalents, mg** | 120 (90, 150) | 135 (105, 165) | 120 (90, 150) |

*P*-values were determined using $\chi^2$ or Fisher's exact tests for categorical variables and analysis of variance or Kruskal–Wallis tests for continuous variables. Categorical data were reported as frequencies (percentages), and continuous variables were reported as medians (quartiles). ACEIs, angiotensin-converting enzyme inhibitors; ARBs, angiotensin II receptor blockers; ASA, American Society of Anesthesiologists; BMI, body mass index; ENT, ear, nose and throat; FAR, fibrinogen to albumin ratio; FPG, fasting plasma glucose; MAP, mean arterial pressure; NLR, neutrophil-lymphocyte ratio; PLR, platelet-to-lymphocyte ratio.

all" strategy assumes all patients receive interventions regardless of risk, while the "treat none" strategy assumes no patient receives interventions. The decision curve evaluates the NB of using the model to selectively treat patients based on predicted risk thresholds compared to these two baseline strategies. In the current model, the NB was 0.002 if all the subjects were treated, and the NB was 0 if none of them were treated. The DCA showed the satisfactory NB that an affected patient could receive from the predictive nomogram, with the predicted probability threshold between 0% and 25% (Fig 1C). The validation dataset (*n* = 67,025) was used to evaluate the model's predictive performance. Internal validation demonstrated consistently strong discrimination with an AUC of 0.878 (95% CI [0.848, 0.909]; *P* < 0.001) (Fig 1D).

## External validation of prediction model

The external validation cohort was comprised of 37,568 patients from Nanfang Hospital and 48,719 patients from Henan Provincial People's Hospital. The model demonstrated strong discrimination in the external validation cohorts, with an AUC of 0.897 (95% CI [0.872, 0.922]; *P* < 0.001) from the Nanfang Hospital data (Fig 2A) and 0.895 (95% CI [0.876, 0.914]; *P* < 0.001) in the Henan Provincial People's Hospital data (Fig 2B). The model exhibited a well-calibrated performance in the cohorts from Nanfang Hospital (Fig 2C) and Henan Provincial People's Hospital (Fig 2D), with overestimation at the extremes in Hosmer and Lemeshow tests, indicating poor calibration in those with higher predicted stroke probability.

## Comparison with existing stroke models

The prediction model herein exhibited improved discrimination with a higher AUC (AUC 0.893, 95% CI [0.879, 0.908]; *P* < 0.001) when compared to the model reported by Mashour and colleagues (AUC 0.777, 95% CI [0.751, 0.803]; *P* < 0.001), the ATRIA stroke risk score (AUC 0.740, 95% CI [0.713, 0.767]; *P* < 0.001), the model of Lip and colleagues (AUC 0.804, 95% CI [0.779, 0.829]; *P* < 0.001), the model of Wolf and colleagues (AUC 0.769, 95% CI [0.744, 0.794]; *P* < 0.001), and the model of Woo and colleagues (AUC 0.883, 95% CI [0.868, 0.899]; *P* < 0.001) (Fig 3A), with a positive NB superior to the five alternative models (Fig 3B). When compared using data from Nanfang Hospital, the prediction model exhibited improved

**Table 2. Variables for perioperative stroke in final multi-variable logistic regression model.**

| Variables | Missing, n (% of 223415) | β coefficient | OR (95% CI) | P-value |
|---|---|---|---|---|
| **ln Age** | 1 (0) | 2.057 | 7.823 (4.599, 13.599) | <0.001 |
| **ASA classification** | 127 (0.1) | | | |
| Class I | | | reference | |
| Class II | | 0.422 | 1.525 (0.861,3.006) | 0.181 |
| Class III | | 0.840 | 2.317 (1.248,4.723) | 0.013 |
| Class IV | | 1.050 | 2.858 (1.361,6.383) | 0.007 |
| **Hypertension** | 2 (0) | 0.496 | 1.642 (1.294,2.084) | <0.001 |
| **Previous stroke** | 2 (0) | 1.851 | 6.369 (4.934,8.178) | <0.001 |
| **Valvular heart disease** | 2 (0) | 1.011 | 2.749 (1.039,5.984) | 0.022 |
| **Preoperative β-blockers** | 0 (0) | 0.486 | 1.626 (1.163,2.234) | 0.003 |
| **Preoperative FPG > 6.1 mmol/L** | 1218 (0.6) | 0.516 | 1.676 (1.315,2.125) | <0.001 |
| **FAR > 0.075** | 3368 (1.5) | 0.370 | 1.448 (1.151,1.828) | 0.002 |
| **Preoperative steroids** | 2 (0) | 0.417 | 1.517 (1.064,2.113) | 0.017 |
| **Preoperative MAP (mmHg)** | 388 (0.2) | 0.016 | 1.016 (1.007,1.025) | <0.001 |
| **Surgery type** | 606 (0.3) | | | |
| ENT | | | reference | |
| Obstetrics and gynecology | | −0.520 | 0.595 (0.281,1.174) | 0.150 |
| Abdominal surgery | | −1.193 | 0.303 (0.191,0.492) | <0.001 |
| Orthopedics | | −0.340 | 0.711 (0.460,1.133) | 0.137 |
| Stomatology | | −0.446 | 0.640 (0.309,1.254) | 0.209 |
| Urology | | −1.087 | 0.337 (0.179,0.618) | 0.001 |
| General surgery | | −14.917 | 0 (0,0) | 0.953 |
| Other | | −1.420 | 0.242 (0.039,0.815) | 0.054 |
| Neurosurgery | | 0.828 | 2.288 (1.499,3.610) | <0.001 |
| Thoracic surgery | | −1.228 | 0.293 (0.142,0.570) | <0.001 |
| Vascular surgery | | −0.809 | 0.445 (0.193,0.939) | 0.043 |
| **Emergency surgery** | 0 (0) | 1.078 | 2.939 (2.012,4.230) | <0.001 |
| **ln (Surgery length)** | 0 (0) | 0.628 | 1.873 (1.524,2.303) | <0.001 |

Age and surgery length were ln transformed. FPG and FAR were transformed to binary data according to the cut-off value. P-values were determined using the Wald test. ASA, American Society of Anesthesiologists; CI, confidence interval; ENT, ear, nose and throat; FAR, fibrinogen to albumin ratio; FPG, fasting plasma glucose; MAP, mean arterial pressure; OR, odds ratio.

discrimination (AUC 0.897, 95% CI [0.872, 0.922]; $P < 0.001$) than the ATRIA stroke risk score (AUC 0.763, 95% CI [0.719, 0.807]; $P < 0.001$) and the model proposed by Lip and colleagues (AUC 0.810, 95% CI [0.769, 0.851]; $P < 0.001$), with a positive NB superior to the two models (S3 Fig). When compared using data from Henan Provincial People's Hospital, the prediction model also exhibited improved discrimination (AUC 0.895, 95% CI [0.876, 0.914]; $P < 0.001$) than the ATRIA stroke risk score (AUC 0.797, 95% CI [0.765, 0.830]; $P < 0.001$) and the model proposed by Lip and colleagues (AUC 0.822, 95% CI [0.792, 0.853]; $P < 0.001$), with a positive NB superior to the two models (S4 Fig).

## Model application

According to the prediction model, we developed an evaluation software named 301PSRC, which can be accessed and applied at the following link: http://139.196.204.68:8087/.

Here, we present two examples of clinical application of our stroke risk calculator. An 82-year-old man with an ASA status of III and a history of stroke was admitted to the hospital

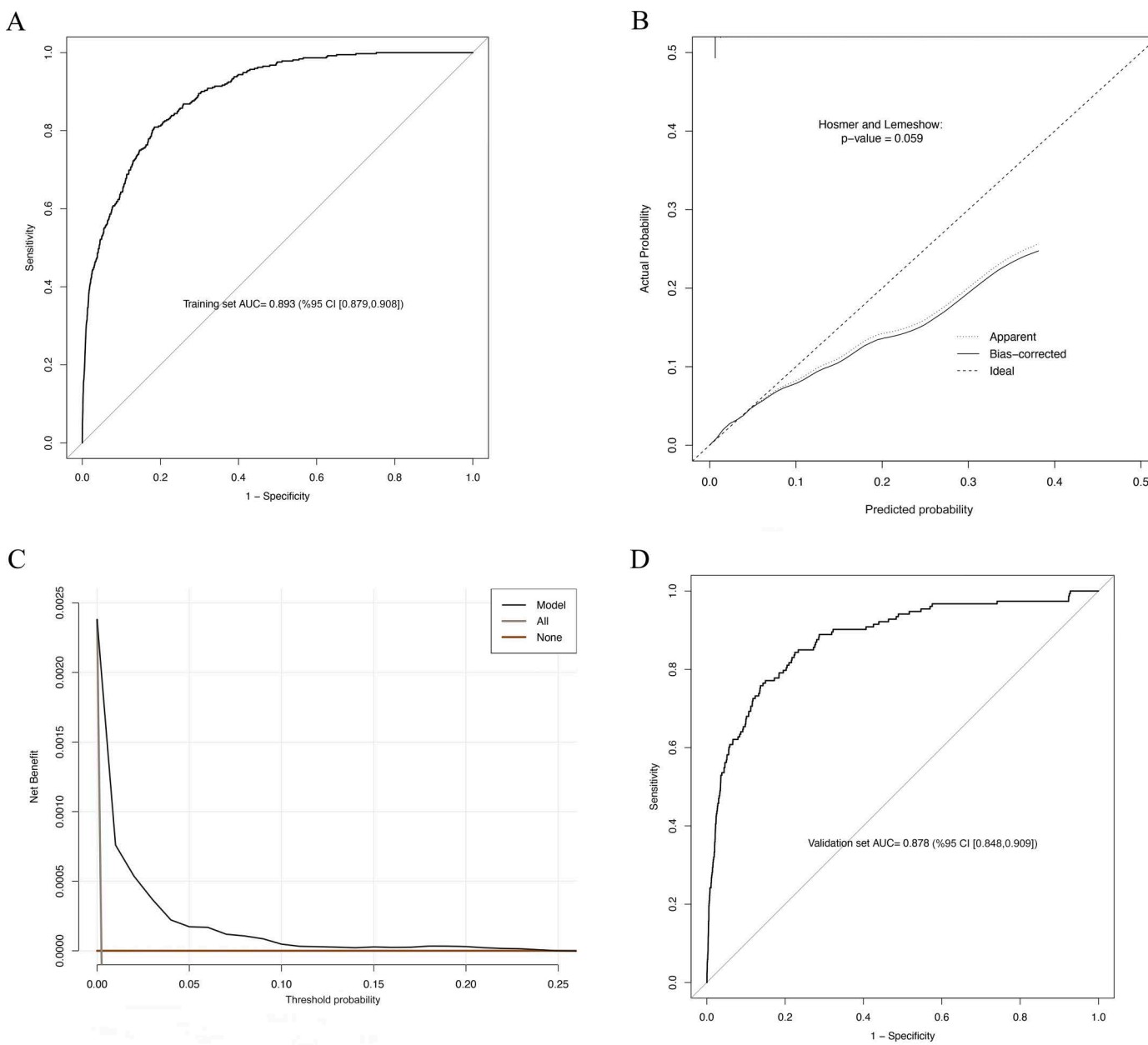

**Fig 1. Development and internal validation of the prediction model.** (A) ROC curve for the training dataset. (B) Calibration curve for the training dataset. *P*-value was determined using Hosmer and Lemeshow test. (C) Decision curve analysis for the training dataset. (D) ROC curve for the internal validation model. AUC, area under the receiver operating characteristic curve. CI, confidence interval. ROC, receiver operating characteristic curve.

for elective abdominal surgery. The preoperative FPG was 7.64 mmol/L, the preoperative MAP was 90, the preoperative fibrinogen was 2.78 g/L, the preoperative serum albumin was 31.5 g/L, and the derived FAR was 0.0882. The surgery duration was 167 min. The patient was identified as having a high risk of perioperative stroke with an estimated absolute risk of 0.6%. A 26-year-old man with an ASA status of II was admitted to the hospital for elective orthopedic surgery. The preoperative FPG was 4.59 mmol/L, the preoperative MAP was 96, the

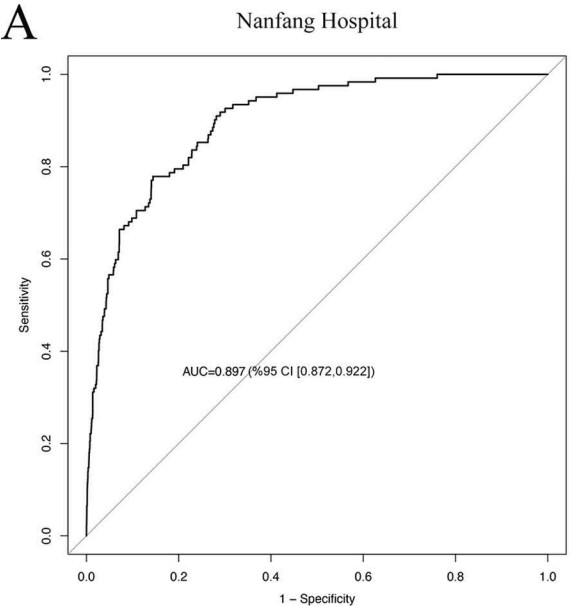

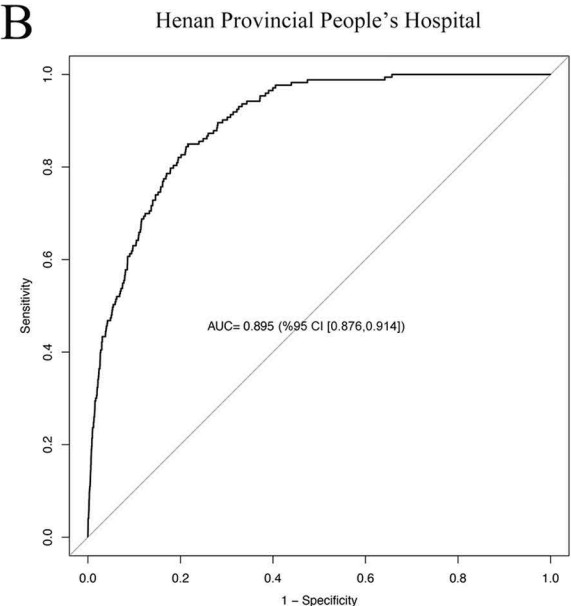

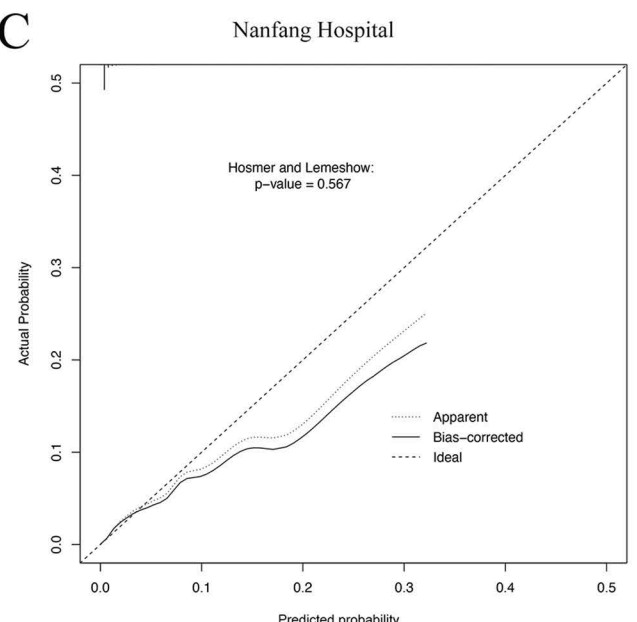

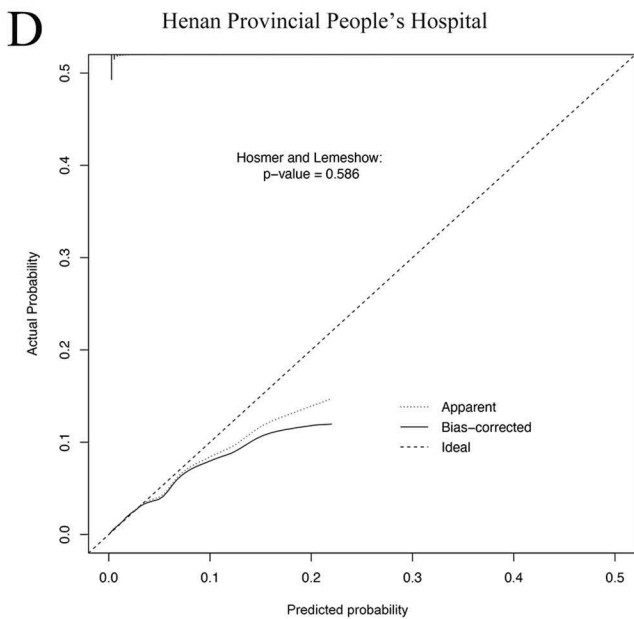

**Fig 2. External validation of the prediction model.** (A) ROC curve for the external validation dataset from Nanfang Hospital. (B) ROC curve for the external validation dataset from Henan Provincial People's Hospital. (C) Calibration curve for the external validation dataset from Nanfang Hospital. *P*-values were determined using Hosmer and Lemeshow tests. (D) Calibration curve for the external validation dataset from Henan Provincial People's Hospital. *P*-value was determined using Hosmer and Lemeshow test. AUC, area under the receiver operating characteristic curve. CI, confidence interval. ROC, receiver operating characteristic curve.

preoperative fibrinogen was 2.67 g/L, the preoperative serum albumin was 38.4 g/L, and the derived FAR was 0.0695. The surgery length was 420 min. The patient was identified as having a low risk of perioperative stroke with an estimated absolute risk of 0.04%.

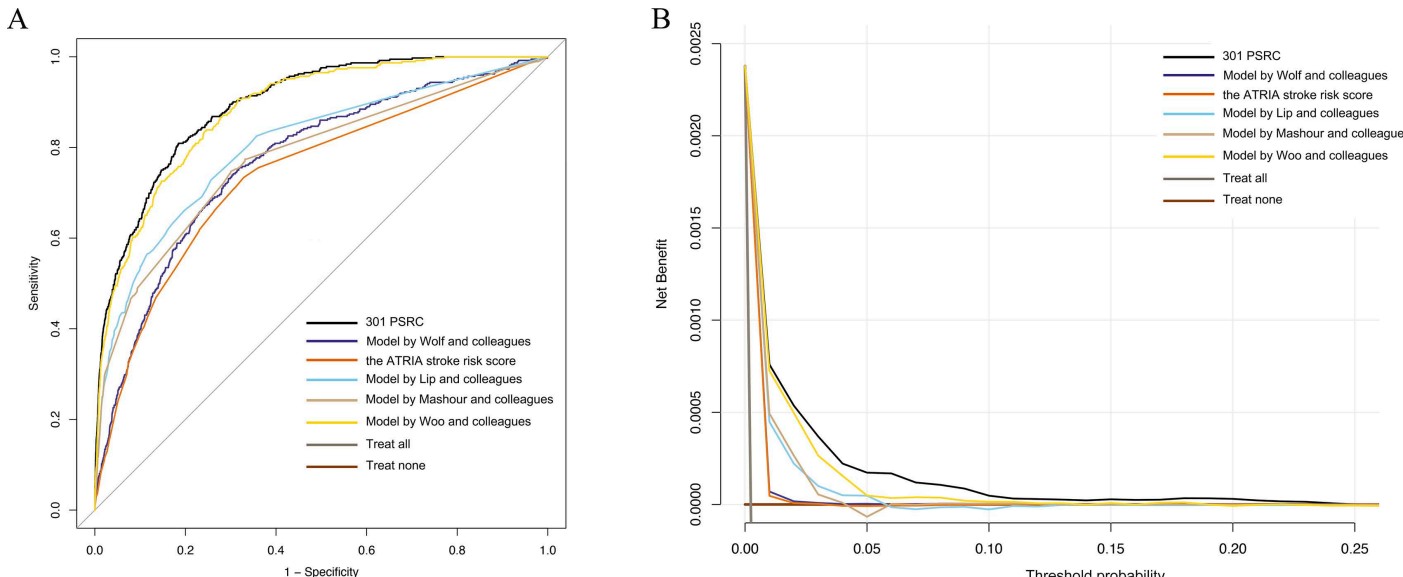

**Fig 3. Comparison with existing models for stroke. (A)** The 301PSRC showed higher AUC when compared to existing models reported by Mashour [5], Lip [15], Wolf [16], Woo [10], and the ATRIA stroke risk score [14]. **(B)** The 301PSRC exhibited a positive net benefit for predicted probability thresholds between 0% and 14%, superior over existing models reported by Mashour, Lip, Wolf, Woo, and the ATRIA stroke risk score.

## Discussion

To our knowledge, our study have identified several new risk factors for perioperative stroke based on a total of 851,383 adult patients undergoing non-cardiac surgery from three independent Chinese medical centers. Our findings were built into a well-validated prediction model that can be used to inform clinicians about patient risk undergoing non-cardiac surgery.

In this study, the incidence of perioperative ischemic stroke in patients undergoing non-cardiac surgery was 0.23%, 0.32%, and 0.36% in three medical centers, respectively. It is worth noting that this is a multi-center study investigating the incidence of perioperative ischemic stroke in patients undergoing non-cardiac surgery in China, which was similar to the incidence found in two recent comprehensive studies from the ACS-NSQIP Database (0.25%) [10] and the VISION study which enlisted data from 28 medical centers across 14 countries (0.3%) [17].

In this study, we identified a total of 31 potential risk factors for perioperative ischemic stroke via univariate analysis. Many of these risk factors have been acknowledged in previous studies such as advanced age, hypertension, renal disease, prior transient ischemic attack/ stroke, myocardial infarction, atrial fibrillation, and diabetes mellitus [4–8]. We also identified several new preoperative risk factors including MAP, NLR, PLR, FAR, FPG, valvular heart disease, and usage of steroids. Finally, several new intraoperative risk factors including crystal liquid dosage, blood product usage, opioid drug dosage (morphine equivalents), and surgical duration further provide new intervention targets for perioperative management to prevent the occurrence of perioperative stroke. However, rigorous cohort studies should be implemented to determine whether these newly identified risk factors are truly associated with perioperative stroke.

We further narrowed down our list of candidate risk factors to 13 independent risk factors using multi-variate analysis including patient age, ASA classification, hypertension, previous

stroke, valvular heart disease, preoperative steroid hormones, preoperative β-blockers, preoperative MAP, preoperative FAR, preoperative FPG, emergency surgery, surgery type, and surgery length, all of which are common clinical indicators with specific clinical significance. As is known, stroke, hypertension and hyperglycemia are significantly age-related diseases [18]. ASA classification, emergency surgery, surgery type, and surgery length are clinical indicators closely related to surgical risk.

Accordingly, we developed a high-performance predictive model for perioperative ischemic stroke with strong discriminative ability (AUC, 0.893). Our model has the highest predictive power for perioperative stroke among previously-developed models such as the ACS-NSQIP Surgical Risk Calculator (AUC, 0.876 and AUC, 0.836) [10,11], the MICA Risk Score (AUC, 0.833) [12], the model reported by Mashour (AUC, 0.773) [5], CHA2DS2-VASc (AUC, 0.744) [9], and Revised Cardiac Risk Index (AUC, 0.743) [19]. The improved power of our model is likely derived from our focus on predicting the incidence of perioperative stroke, whereas the aforementioned prediction models considered perioperative stroke as one of the numerous outcomes. More importantly, our model was validated against data from two other medical centers with high discrimination (AUC, 0.897 and AUC, 0.895, respectively), which indicated the well applicability of our model to the general Chinese population. Since the sample size of individuals with higher stroke probability was small, both external validation cohorts exhibited poor calibration in those with higher predicted stroke probability, thus, the calibration of the prediction model is still open to debate.

We incorporated five existing models [5,10,14–16] into our database for perioperative stroke risk prediction without the ACS-NSQIP Surgical Risk Calculator [11] and MICA Risk Score [12] because of the discrepancy among their main outcome. Furthermore, performance between the CHA2DSVASc score [9] and the 301PSRC were not compared either, as the variable "congestive heart failure" used in the CHA2DSVASc score could not be identified (the category of heart failure were not specified) in the current database. Notably, our model still had the highest discriminative ability for perioperative stroke compared to the four existing models. Taken together, our model maximizes the clinical benefits for patients compared to existing models and should be promoted as the effective model of perioperative stroke risk prediction based on our study.

In our opinion, our model was built upon several notable strengths. First, we constructed a large dataset from 376,933 non-cardiac surgical inpatients, from which we developed an innovative prognostic model specific for perioperative stroke with the highest discriminative power. To the best of our knowledge, this is the first large-scale patient database targeting perioperative stroke in China. A second major strength is the external validation from two other independent medical centers from 37,568 patients and 48,719 patients which further refined and supported our efforts. Third, we developed an online prognostic software platform with intervention measures so that this model can immediately be incorporated into the common medical protocol and benefit clinicians and patients alike.

Despite the advantages of our study, we recognize that there are some limitations to this work. Firstly, we excluded surgical patients with an operation duration of less than one hour, which might limit the application of this prediction model, particularly for less-intensive surgeries. However, the incidence rate of perioperative stroke in these patients was extremely low (0.077%, 45/58401), which is of little predictive value and might have limited effectiveness of our model which we intended to address the population most vulnerable to perioperative stroke. Secondly, the construction and external validation of our model were from three independent retrospective databases within China, which lack validation from prospective databases and non-Chinese databases. In the future, we aspire to cooperate with more hospitals to

establish prospective databases, screen patients with perioperative stroke, and integrate these data into our model to consistently optimize and verify our prediction model.

In conclusion, we identified 13 independent risk factors for perioperative stroke, which provided particular insight into intraoperative indicators, and constructed an effective and externally validated model to predict the incidence of perioperative stroke within Chinese patients undergoing non-cardiac surgery. Our results provide potential intervention targets and screen high-risk patients for perioperative stroke prevention.

## Supporting information

**S1 Fig. Study flow chart.** ASA, American Society of Anesthesiologists.
(TIF)

**S2 Fig. Nomogram for predicting the probability of perioperative stroke.** Age and surgery length were ln transformed. FPG and FAR were transformed to binary data according to the cut-off values. For binary data, 0 represents "no" and 1 represents "yes". ASA, American Society of Anesthesiologists; MAP, mean arterial pressure; FAR, fibrinogen to albumin ratio; FPG, fasting plasma glucose. Surgery type 1, ear, nose, and throat; 2, obstetrics and gynecology; 3, abdominal surgery; 4, orthopedics; 5, stomatology; 6, urology; 7, general surgery; 8, other surgeries; 9, neurosurgery; 10, thoracic surgery; 11, vascular surgery.
(TIF)

**S3 Fig. Comparison with existing models on the data of Nanfang Hospital.** (**A**) The 301 PSRC showed higher AUC when compared to the model reported by Lip [15] and the ATRIA stroke risk score [14]. (**B**) The 301 PSRC exhibited a positive net benefit superior over the model reported by Lip and the ATRIA stroke risk score.
(TIF)

**S4 Fig. Comparison with existing models on the data of Henan Provincial People's Hospital.** (**A**) The 301 PSRC showed higher AUC when compared to the model reported by Lip [15] and the ATRIA stroke risk score [14]. (**B**) The 301 PSRC exhibited a positive net benefit superior over the model reported by Lip and the ATRIA stroke risk score.
(TIF)

**S1 Table. TRIPOD checklist: prediction model development and validation.**
(DOCX)

**S2 Table. ICD-9/10 diagnosis codes for ischemic stroke.**
(DOCX)

**S3 Table. Patient characteristics in the validation cohorts.**
(DOCX)

**S4 Table. Univariate analysis.**
(DOC)

**S5 Table. Details for the unadjusted analyses.**
(DOC)

**S6 Table. Details of rejected variables.**
(DOC)

**S7 Table. Scales for the 13 predictor variables.**
(DOC)

## Acknowledgments

We would like to thank Wei Wei, Lan Sun, and Tongyan Sun of Hangzhou Le9 Healthcare Technology Co., Ltd., for their assistance in the clinical data extraction for this study.

## Author contributions

**Conceptualization:** Yulong Ma, Siyuan Liu, Faqiang Zhang, Xuhui Cong, Bingcheng Zhao, Kexuan Liu, Jiaqiang Zhang, Weidong Mi.

**Data curation:** Yulong Ma, Siyuan Liu, Faqiang Zhang, Xuhui Cong, Bingcheng Zhao, Miao Sun, Huikai Yang, Min Liu, Peng Li, Yuxiang Song, Jiangbei Cao, Yingfu Li, Wei Zhang.

**Formal analysis:** Yulong Ma, Siyuan Liu, Faqiang Zhang, Xuhui Cong, Bingcheng Zhao, Miao Sun, Huikai Yang, Min Liu, Peng Li, Yuxiang Song, Jiangbei Cao, Yingfu Li, Wei Zhang.

**Investigation:** Yulong Ma, Siyuan Liu, Faqiang Zhang, Xuhui Cong, Bingcheng Zhao, Miao Sun, Huikai Yang, Min Liu, Peng Li, Yuxiang Song, Jiangbei Cao, Yingfu Li, Wei Zhang.

**Methodology:** Yulong Ma, Siyuan Liu, Faqiang Zhang, Xuhui Cong, Bingcheng Zhao, Kexuan Liu, Jiaqiang Zhang, Weidong Mi.

**Project administration:** Kexuan Liu, Jiaqiang Zhang, Weidong Mi.

**Resources:** Yulong Ma, Siyuan Liu, Faqiang Zhang, Xuhui Cong, Bingcheng Zhao, Miao Sun, Huikai Yang, Min Liu, Peng Li, Yuxiang Song, Jiangbei Cao, Yingfu Li, Wei Zhang.

**Software:** Yulong Ma, Siyuan Liu, Faqiang Zhang, Xuhui Cong, Bingcheng Zhao, Miao Sun, Huikai Yang, Min Liu, Peng Li, Yuxiang Song, Jiangbei Cao, Yingfu Li, Wei Zhang.

**Supervision:** Kexuan Liu, Jiaqiang Zhang, Weidong Mi.

**Validation:** Yulong Ma, Siyuan Liu, Faqiang Zhang, Xuhui Cong, Bingcheng Zhao, Miao Sun, Huikai Yang, Min Liu, Peng Li, Yuxiang Song, Jiangbei Cao, Yingfu Li, Wei Zhang, Kexuan Liu, Jiaqiang Zhang, Weidong Mi.

**Writing – original draft:** Yulong Ma, Siyuan Liu, Faqiang Zhang, Xuhui Cong, Bingcheng Zhao.

**Writing – review & editing:** Kexuan Liu, Jiaqiang Zhang, Weidong Mi.

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
