## [Editor Report · Decision Letter 0]

21 Dec 2023

Dear Dr Ma, 

Thank you for submitting your manuscript entitled "Risk Factor Analysis and Creation of an Externally-Validated Prediction Model for Perioperative Stroke Following Non-Cardiac Surgery" for consideration by PLOS Medicine.

Your manuscript has now been evaluated by the PLOS Medicine editorial staff and I am writing to let you know that we would like to send your submission out for external peer review.

Please re-submit your manuscript within two working days, i.e. by Dec 25 2023 11:59PM.

Kind regards,

Alexandra Schaefer, PhD

Associate Editor

PLOS Medicine

---

## [Decision Letter · Decision Letter 1]

30 Jan 2024

Dear Dr. Ma,

Thank you very much for submitting your manuscript "Risk Factor Analysis and Creation of an Externally-Validated Prediction Model for Perioperative Stroke Following Non-Cardiac Surgery" (PMEDICINE-D-23-03773R1) for consideration at PLOS Medicine. 

Your paper was evaluated by an associate editor and discussed among all the editors here. It was also discussed with an academic editor with relevant expertise, and sent to independent reviewers, including a statistical reviewer. The reviews are appended at the bottom of this email and any accompanying reviewer attachments can be seen via the link below:

[LINK]

In light of these reviews, I am afraid that we will not be able to accept the manuscript for publication in the journal in its current form, but we would like to consider a revised version that addresses the reviewers' and editors' comments. Obviously we cannot make any decision about publication until we have seen the revised manuscript and your response, and we plan to seek re-review by one or more of the reviewers. 

Please use the following link to submit the revised manuscript: https://www.editorialmanager.com/pmedicine/

We expect to receive your revised manuscript by Feb 20 2024. However, if this deadline is not feasible, please contact me by email, and we can discuss a suitable alternative.

Don’t hesitate to contact me directly with any questions (aschaefer@plos.org). If you reply directly to this message, please be sure to ‘Reply All’ so your message comes directly to my inbox.

We look forward to receiving your revised manuscript.

Sincerely,

Alexandra Schaefer, PhD

PLOS Medicine

plosmedicine.org

ACADEMIC EDITOR COMMENTS

Many of the comments are important but addressable. I think the following are critical: 

1) Comment #1 (‘perioperative aspirin’) by Reviewer 4 - very important and needs a clear response

2) Comment #9, #10 and #11 by Reviewer 2

EDITORIAL COMMENTS

In line with the reviewers' comments, we strongly feel that validation in a non-Chinese cohort would be ideal, but are not going to make this a mandatory requirement. Please consider whether including the final model variables in the abstract would be beneficial without overloading the structure of the abstract.

FINCANCIAL DISCLOSURE

The funding statement should include: specific grant numbers, initials of authors who received each award, URLs to sponsors’ websites. Also, please state whether any sponsors or funders (other than the named authors) played any role in study design, data collection and analysis, the decision to publish, or preparation of the manuscript. If they had no role in the research, include this sentence: “The funders had no role in study design, data collection and analysis, decision to publish, or preparation of the manuscript.”

COMEPTING INTEREST

All authors must declare their relevant competing interests per the PLOS policy, which can be seen here: https://journals.plos.org/plosmedicine/s/competing-interests

For authors with ties to industry, please indicate whether any of the interests has a financial stake in the results of the current study.

GENERAL COMMENTS

1) Please include page numbers and line numbers in the manuscript file. Use continuous line numbers (do not restart the numbering on each page). For review purposes, we started counting the Abstract as page 1.

2) Please cite the reference numbers in square brackets (e.g., “We used the techniques developed by our colleagues [19] to analyze the data”). Citations should be preceding punctuation.

3) Please cite your Supporting Information as outlined here: https://journals.plos.org/plosmedicine/s/supporting-information

4) Please ensure that the study is reported according to the TRIPOD guideline and include the completed TRIPOD checklist as Supporting Information. When completing the checklist, please use section and paragraph numbers, rather than page numbers. Please add the following statement, or similar, to the Methods: "This study is reported as per the Transparent reporting of a multivariable prediction model for individual prognosis or diagnosis (TRIPOD): The TRIPOD statement (S1 Checklist).”

ABSTRACT

1) PLOS Medicine requests that main results are quantified with 95% CIs as well as p values. When reporting p values please report as p<0.001 and where higher as the exact p value p=0.002, for example. For the purposes of transparent data reporting, if not including the aforementioned please clearly state the reasons why not.

2) Throughout, suggest reporting statistical information as follows to improve clarity for the reader “22% (95% CI [13%,28%]; p</=)”. Please amend throughout the abstract and main manuscript. Please note the use of commas to separate upper and lower bounds, as opposed to hyphens as these can be confused with reporting of negative values.

3) When a p value is given, please specify the statistical test used to determine it. 

4) Please structure your abstract using the PLOS Medicine headings (Background, Methods and Findings, Conclusions). Please combine the Methods and Findings sections into one section, “Methods and findings”.

5) Please ensure that all numbers presented in the abstract are present and identical to numbers presented in the main manuscript text.

6) Please include the age of the study population.

7) Please include the important dependent variables that are adjusted for in the analyses.

8) In the last sentence of the Abstract Methods and Findings section, please describe the main limitation(s) of the study's methodology.

9) Abstract Conclusions:

* Please interpret the study based on the results presented in the abstract, emphasizing what is new without overstating your conclusions.

* Please avoid assertions of primacy ("novel...."). Please temper claims of primacy of results by stating, "to our knowledge" or something similar.

10) Please ensure that the abstract is reported according to the TRIPOD guideline.

AUTHOR SUMMARY

At this stage, we ask that you include a short, non-technical Author Summary of your research to make findings accessible to a wide audience that includes both scientists and non-scientists. The Author Summary should immediately follow the Abstract in your revised manuscript. This text is subject to editorial change and should be distinct from the scientific abstract. Please see our author guidelines for more information: https://journals.plos.org/plosmedicine/s/revising-your-manuscript#loc-author-summary.

The summary should include 2-3 single sentence, individual bullet points under each of the questions. The last bullet under ‘What Do These Findings Mean?’ point should describe the main limitation of the study's methodology.

It may be helpful to review currently published articles for examples which can be found on our website here https://journals.plos.org/plosmedicine/

INTRODUCTION

If there has been a systematic review of the evidence related to your study (or you have conducted one), please refer to and reference that review and indicate whether it supports the need for your study.

METHODS AND RESULTS

1) PLOS Medicine requests that main results are quantified with 95% CIs as well as p values. When reporting p values please report as p<0.001 and where higher as the exact p value p=0.002, for example. For the purposes of transparent data reporting, if not including the aforementioned please clearly state the reasons why not. Please include any important dependent variables that are adjusted for in the analyses. We suggest reporting statistical information as detailed above – see under ABSTRACT.

2) Please present numerators and denominators for percentages, at least in the Tables [not necessarily each time they're mentioned].

3) The link for the 301 Perioperative Stroke Risk Calculator (301PSRC) cannot be accessed. Please check.

4) Please report the Methods and Results according to the TRIPOD Checklist: Prediction Model Development and Validation.

DISCUSSION

Please ensure to present and organize the Discussion as follows: a short, clear summary of the article's findings; what the study adds to existing research and where and why the results may differ from previous research; strengths and limitations of the study; implications and next steps for research, clinical practice, and/or public policy; one-paragraph conclusion. Please avoid the use of subheadings such that the discussion reads as continuous prose.

1) Please temper claims of primacy of results by stating, "to our knowledge" or something similar.

(e.g. "We also discovered several new preoperative risk factors...."). 

2) Please temper claims of “preeminent”, “excellent” and the like.

TABLES

1) Throughout please ensure that each table is affiliated to an appropriate title and caption which clearly describes the table content without the need to refer to the text.

2) Please ensure that all abbreviations including those used for statistical reporting are clearly defined in the caption or footnote.

3) Please ensure that where confidence intervals are reported, p values are also reported as described above. Please note the use of commas to separate upper and lower bounds, as opposed to hyphens as these can be confused with reporting of negative values.

FIGURES

1) For all Figures, please ensure that you have complied with our figures requirements http://journals.plos.org/plosmedicine/s/figures.

2) Please consider avoiding the use of red and green in order to make your figure more accessible to those with colour blindness. 

3) Please in the figure legend/description, define abbreviations used in each figure (including those in Supporting Information files).

4) Please provide titles, legends and descriptions for all figures (including those in Supporting Information files).

5) Figure 1: The graphs are rather small and the details are not easy to read. Please revise.

6) Figure 1: In Figure 1 A, please explain the different scales for the different factors. What does 'In age' mean? Please remove 'Lorem Ipsum' from Graph D. In Graph D, the different colors, especially the two gray colors, are difficult to distinguish - please revise.

7) Figure 3: Please provide references for the models being compared.

SUPPLEMENTARY MATERIAL

1) For supplementary figures and tables, please see the general comments under TABLES and FIGURES (color, abbreviations, titles, descriptions, etc.) and amend accordingly.

2) We suggest reporting statistical information as detailed above – see under ABSTRACT. Please define all numerical values.

3) As for the main manuscript, please indicate whether analyses are adjusted to help facilitate transparent data reporting please also detail the factors adjusted for and present the unadjusted analyses for comparison. If not, please clearly state the reasons why not.

REFERENCES

1) PLOS uses the numbered citation (citation-sequence) method and first six authors, et al.

2) Please ensure that journal name abbreviations match those found in the National Center for Biotechnology Information (NCBI) databases (http://www.ncbi.nlm.nih.gov/nlmcatalog/journals), and are appropriately formatted and capitalised.

3) Where website addresses are cited, please specify the date of access. 

4) Please also see https://journals.plos.org/plosmedicine/s/submission-guidelines#loc-references for further details on reference formatting. 

Comments from the reviewers:

Reviewer #1: The author identified the risk factors and developed prediction model for perioperative stroke following non-cardiac surgery. This study could be expected to have an important impact on daily clinical management for prevention of perioperative stroke . However, there are some questions and comments about contents of the manuscript.

In the Introduction section, there is no need to explain what "ischemic stroke" means.

The description of the 301 PSRC was in the last paragraph in the result section, but the analysis using this software was described in the preceding 6th paragraph and figure 3. The author should reconsider the order of the contents about the 301 PSRC. 

As the authors mentioned, it would be useful to focus on intraoperative factors to reduce the risk of perioperative stroke and to detect perioperative stroke onset earlier. However, if the reason for the higher predictive ability of this model compared to previous models is due to the employment of intraoperative factors in this study, then it is not surprising that this model has a higher predictive ability. For the purpose of preoperative stroke risk prediction only, a model consisting only of information available before the start of surgery is favorable. The authors should mention these points as limitations of this study.

Reviewer #2: "Risk Factor Analysis and Creation of an Externally-Validated Prediction Model for Perioperative Stroke Following Non-Cardiac Surgery" describes the development and (external) validation of a risk (nomogram) model, for the prediction of perioperative stroke risk. The model was developed on a cohort of some 223,000 non-cardiac surgical patients from 2008-2019 attending the First Medical Center of Chinese PLA General Hospital, and externally validation on data from two other Chinese hospitals. An AUC of about 0.90 was reported on all three cohorts.

While the scale of the study is impressive, some issues might be considered:

1. In the Abstract, AUC values were given to various precisions (i.e. 0.91, 0.9, 0.894). The number of decimal places/significant figures might be standardized here and throughout the manuscript if appropriate. The 95% CIs for the external validation cohort AUCs might also be stated.

2. In the Study design and cohorts section, various exclusion criteria were described. A flowchart showing the number of patients eliminated according to each of these criteria might be included as a figure if possible.

3. In the Study design and cohorts section, "missing data for any variables" was stated as an exclusion criteria. However, imputation for variables with <20% missing data is later mentioned, which implies that patients with missing data for some variables were included. This might be clarified.

4. In the Predictor variables section, it is stated that multiple imputation was used for variables with <20% missing data. Details of this multiple imputation, including the method (e.g. MCMC, single value, etc.) and any parameters might be briefly mentioned.

5. In the Model development and validation section, it is stated that stepwise elimination was used to select the final 14 variables. The p-values of rejected variables might be included in a table, possibly in supplementary material.

6. In the Statistical analyses section, external data was stated to be collected from Nanfang and Henan. It might be stated whether these external cohorts followed the same exclusion criteria as the development cohort.

7. The missing data percentage for each variable might be included in Table 2, if possible.

8. In the Prediction Model for Perioperative Stroke section, a nomogram (Figure 1A) was introduced, and a cutoff of 158.3 points stated. The construction of this nomogram (and cutoff, including specific definition of "high-risk" - e.g. percentage chance of developing stroke within 30 days) might be briefly described/referenced, possibly in supplementary material.

9. In the Comparison with existing stroke models section, it is stated that the developed model exhibited improved discrimination (AUC=0.91) compared to a number of existing stroke models. However, this appears to be analyzed only on the developmental cohort, which might present the developed model with an advantage (since it was developed on that cohort itself). As such, it would be strongly recommended to also compare the developed model against existing stroke models, on the external validation cohorts.

10. Moreover, it is not clear that the four models compared against (Wolf et al., ATRIA, Lip et al., Mashour et al.) are the most appropriate/relevant, partly as they are fairly dated (with ATRIA being the newest, from 2013). It might thus also be strongly recommended to include newer models (e.g. "Cardiovascular Risk Scores to Predict Perioperative Stroke in Noncardiac Surgery", Wilcox et al., Stroke 2019; "Development and Validation of a Prediction Model for Stroke, Cardiac, and Mortality Risk After Non‐Cardiac Surgery", Woo et al., JAHA 2021), and also the most commonly-used models such as CHA2DSVASc, ACS-NSQIP, MICA etc., especially since some of their AUCs are given in the Discussion section, but are not analyzed in Figure 3 and the relevant results section) in the comparison.

11. Additionally, the external validation data is also derived exclusively from Chinese populations (Nanfang/Henan). If possible, it should be considered to also externally validate on non-Chinese/international datasets, to determine whether the performance improvement is specific to ethnicity.

12. In the Practical Application of Risk Model section, a link to the 301PSRC online calculator was given as http://211.166.249.201:6003. This link however does not appear to display the risk calculator. This might be addressed.

13. In the Discussion section, it is stated that "We incorporated four existing models into our database for perioperative stroke risk prediction, without the ACS-NSQIP Surgical Risk Calculator and MICA Risk Score as the discrepancy of the indicators". This sentence might be clarified.

14. In Table 2, the actual variables might be bolded for clarity, with the categories within variables (e.g. for Surgery Type) left unbolded and possibly offset.

15. In Table 2, "Emergent surgery" might be "Emergency surgery".

16. The chart title for Figure 1D (Decision Curve Analysis) is "Lorem Ipsum" placeholder text. This might be replaced with the actual intended title.

Reviewer #3: The present research is clinically relevant, thank you for letting me peer review it. The following two points are my comments.

1. I think it is important for authors to have developed an application that is easy to use, and I think it is expected that this risk model will be used before non-cardiac surgery especially in China through the newly developed application. However, I could not identify the application from the link presented in the manuscript. Could you correct this so that I can see what that application is?

2. The exclusion criteria include patients with missing data for any variables, but the Predictor Variables subsection states that: 

"Variables with ≥20% missing data were excluded. We used multiple imputations to impute missing values for variables with <20% missing data", 

which is inconsistent.

I could not understand whether this modeling study was based on complete data analysis or imputed data analysis.

Reviewer #4: Yulong Ma and colleagues reported methods and results of the development of a prediction model for the risk of perioperative stroke within 30 days after noncardiac surgery, analyzing administrative hospital data from three Chinese hospitals. Data from adult patients that underwent noncardiac surgery from 2008-2019 at The First Medical Center of Chinese PLA General Hospital were used as derivation cohort; the model was then externally validated using the hospital records of two other Chinese centres. They also compared the predictive performance of their model with that of other existing prediction models. They also translated their model into a risk calculator for clinical use.

The study itself could add to our understanding of risk factors for perioperative stroke after noncardiac surgery and could eventually offer a prediction tool that, after further validation (prospectively and externally in other countries), could even enter clinical practice. The statistical methods the investigators used appear quite rigorous. There is also to acknowledge that, among existing cohort studies with multivariable analyses of perioperative stroke predictors, the databases here analyzed would provide the most recent data (see Marcucci et al., Lancet Neurol 2023; 22: 946-58). There are however major concerns with the study; in particular one possible major flow that I recommend to address before considering the article for publication.

Major concern

1. Among the variables they consider as possible risk factors, the authors include what they name "perioperative aspirin". The only definition they provide for this is that "the patient received aspirin medication perioperatively." What "perioperatively" mean? If this means (as usually) that they considered when patients took aspirin any time around surgery, inclduing pre-, potentially intra-, and post-operatively (for example on any day after surgery they spent in hospital, or even only on the first few days after surgery), this means this could be aspirin given BECAUSE the patient had an intra- or postoperative stroke. This means, that to include perioperative aspirin as a risk factor or predictor for stroke is reverse association and therefore biased analysis. If this is what they did, the very strong association between perioperative aspirin and stroke shown in the univariable and multivariable models is then easily explained; as explained is the outstanding discrimination performance (AUC>0.9) the authors found for their model (exceptional for a prognostic model). If this is what they did, the authors should remove this variable from the studied risk factors, and re-run the multivariable analyses, and provide a new model and risk calculator.

If, instead, "perioperative aspirin" had a different meaning (for example, it is preoperative aspirin) this should be amended or clarified. 

Other major points

2. The rationale for selecting the models they selected for comparison, and not other existing ones, should be better explained. Only one of the selected models was developed to predict perioperative stroke; the other ones were models for non-operative stroke; two of them for the risk of stroke in patients with atrial fibrillation. For instance, it is unclear why the authors did not compare their model with that of Woo et al. (reference 10), which is a prediction model developed in a similar way as the one here developed but from an US hospital database (i.e., the comparison would have been more relevant and interesting). In the introduction the authors mention Woo et al. among the existing models which have the limitation of predicting "various perioperative complications of the cerebrovascular system" at once; in fact, Woo et al. does include also a specific prediction model for stroke. If not selected as a comparator model, the authors could at least discuss their findings compared to those of Woo et al.'s findings.

On this point, the sentence they include in the Discussion ("We incorporated four existing models,13-15 into our database for perioperative stroke risk prediction, without the ACS-NSQIP Surgical Risk Calculator11 and MICA Risk Score12 as the discrepancy of the indicators") is not clear, and would require rephrasing or explanation. 

3. If the model confirms its very good performance even once point 1 is addressed, still, the authors should recommend caution in the use of their model and risk calculator. This remains a model developed retrospectively from administrative hospital databases. Furthermore, the fact that the model was developed and externally validated only in a Chinese population and hospital system might eventually prove irrelevant, but we will not know it until it is externally validated in different countries/hospital systems. The authors did acknowledge the lack of prospective validation among the study limitations. My recommendation would be to mitigate the call for use of their model and online calculator including considerations of these limitations in the Conclusions in the abstract and in the text.

4. The definition of the timing of their outcome should be more accurate. Did the "within 30 days following surgery" include the day of surgery?

5. To include a Decision Curve Analysis seems a very interesting and possibly helpful approach. The authors should better explain what they did (even including some of the more detailed methods in the online supplementary material) and how the reader should interpret the results.

Minor comments

6. Regarding the sentence in the Introduction: "Unfortunately, most risk factors to date entail

preoperative indicators due to limited accessibility to intra-operative indicators." I would not necessarily put a lot of emphasis on intraoperative predictors since they might help refining the prediction but at the methodological cost of possibly introducing a bias. Some of the perioperative strokes are occurring during the procedure (while the patient is sedated) and what we include as a predictor (for example intraoperative hemodynamics) could in fact be a consequence of or expression of the stroke that is happening. This is particularly possible when the prediction model is based on administrative databases (and no actual date of the event can be used in the analysis) and not based on prospectively collected data where the time relationship between exposure and outcome can be taken into account. 

7. In the evaluation of prognostic models, what the Hosmer-Lemeshow test assesses is typically called 'calibration' (as opposed to discrimination) and not "model accuracy".

8. Table 1 is very difficult to read. The authors (or editors) should consider to change the formatting. One way would be to have in this Table 1 only the description of the derivation (and internal validation) cohort; and have a similar supplementary table for the characteristics of the other two cohorts.

9. Table 1 is introduced in the text as if it is showing only the patient "demographics", but this is not the case.

[LINK]

---

## [Decision Letter · Decision Letter 2]

29 May 2024

Dear Dr. Ma,

Thank you very much for submitting your manuscript "Risk Factor Analysis and Creation of an Externally-Validated Prediction Model for Perioperative Stroke Following Non-Cardiac Surgery" (PMEDICINE-D-23-03773R2) for consideration at PLOS Medicine. 

Thank you for your detailed response to the editors' and reviewers' comments. I have discussed the paper with my colleagues and the academic editor, and it has also been seen again by two of the original reviewers. The changes made to the paper were mostly satisfactory to the reviewers. However, the editorial team concurs with reviewer #4 that their comments regarding the inclusion of perioperative aspirin have not been sufficiently addressed. Therefore, we ask you to carefully address the comments in a further revision and suggest that you provide the data collection forms, which should indicate that you were able to differentiate between pre- and post-operative aspirin, to preclude the need for further revisions and satisfy the reviewers and editors. When submitting your revised paper, please again include a detailed point-by-point response to the comments.

The reviews are appended at the bottom of this email and any accompanying reviewer attachments can be seen via the link below:

[LINK]

In light of these reviews, I am afraid that we will not be able to accept the manuscript for publication in the journal in its current form, but we would like to consider a revised version that addresses the reviewers' and editors' comments. Obviously we cannot make any decision about publication until we have seen the revised manuscript and your response, and we plan to seek re-review by one or more of the reviewers. 

Please use the following link to submit the revised manuscript: https://www.editorialmanager.com/pmedicine/

We expect to receive your revised manuscript by Jun 19 2024. However, if this deadline is not feasible, please contact me by email, and we can discuss a suitable alternative.

Don't hesitate to contact me directly with any questions (atosun@plos.org). If you reply directly to this message, please be sure to 'Reply All' so your message comes directly to my inbox.

We look forward to receiving your revised manuscript. 

Sincerely,

Alexandra Tosun, PhD

PLOS Medicine

plosmedicine.org

Comments from the reviewers:

Reviewer #2: We thank the authors for largely addressing our previous concerns; it appears that the exclusions were already included in Supplementary Figure 1 in the original submission - apologies for the oversight.

A couple of comments remain:

1. A minor suggestion would be to again standardize the decimal points for percentages in Table 2, if possible.

2. It is stated that "We have stated the reason of not comparising the CHA2DSVASc, ACSNSQIP and MICA were stated in the Discussion section in page 19". However, it is unclear if CHA2DSVASc was discussed (in the paragraph from Line 379 to 385). This might be clarified.

Reviewer #4: It is appreciable how the authors tried to address all the reviewers' comments and modified the manuscript accordingly. The manuscript is overall improved. 

However, I do remain very concerned regarding the inclusion of perioperative aspirin among their predictors (which, not surprisingly, turned out to be among the strongest ones).

My initial comment was: 

"1. Among the variables they consider as possible risk factors, the authors include what they name "perioperative aspirin". The only definition they provide for this is that "the patient received aspirin medication perioperatively." What "perioperatively" mean? If this means (as usually) that they considered when patients took aspirin any time around surgery, inclduing pre-, potentially intra-, and post-operatively (for example on any day after surgery they spent in hospital, or even only on the first few days after surgery), this means this could be aspirin given BECAUSE the patient had an intra- or postoperative stroke. This means, that to include perioperative aspirin as a risk factor or predictor for stroke is reverse association and therefore biased analysis. If this is what they did, the very strong association between perioperative aspirin and stroke shown in the univariable and multivariable models is then easily explained; as explained is the outstanding discrimination performance (AUC>0.9) the authors found for their model (exceptional for a prognostic model). If this is what they did, the authors should remove this variable from the studied risk factors, and re-run the multivariable analyses, and provide a new model and risk calculator.

If, instead, "perioperative aspirin" had a different meaning (for example, it is preoperative aspirin) this should be amended or clarified."

The authors' response to my comment is:

"Response: We thank the reviewer for this crucial question, which we had not elaborated

clearly in the Methods. We apologize for the confusion caused by our unclear description. "The patient received aspirin medication perioperatively" means that the patients took aspirin both pre-operatively and post-operatively for anti-coagulation medication, not because the patient had an intra- or postoperative stroke. We have clarified this variable in the revised Methods in page 8 in line 161-162."

They then addressed this point is "Perioperative aspirin indicated the patients who took aspirin both pre-operatively and post-operatively for anti-coagulation medication."

Beyond including an erroneous statement (aspirin is not an anticoagulant), this revision does not address the fundamentals of my concern.

The fact that the patient was already on aspirin preoperatively and they were restarted on their aspirin postoperatively, does not change the concern that the choice of restarting aspirin could have been affected by the occurrence of stroke, and the risk of reverse causation still exists. This risk would not exist if the authors were able to include in their model only aspirin use that preceded the day of occurrence of stroke - which is a very improbable thing that they could do, since rarely administrative databases allow to account for this type of precise temporal relationships.

Even if we were to assume that they were able to account for perioperative aspirin use that only preceded the occurrence of stroke, the inclusion of perioperative aspirin in a postoperative stroke prediction model remains both not appropriate and possibly not helpful, as I try to explain with the following points:

- the most probable way here perioperative aspirin is a risk factor (if not for, or in addition to, reverse causation) is being a surrogate (or confounder) for high cardiovascular risk (in addition to the cardiovascular risk predictors they have already included). For example, patients with a history of coronary stenting are those who could have most typically aspirin continued perioperatively, and they could have an increased risk of stroke. To try to have in the model those predictors that in fact explained why aspirin was continued perioperatively would be way more meaningful to a physician reading the article and using the model, than having periop aspirin as a predictor; indeed,

- what could physicians do with this knowledge that, in this observation administrative database, perioperative aspirin continuation was strongly associated with an increased risk of postoperative stroke? When they run this model preoperatively, or even immediately postoperatively, would they decide not to continue aspirin? Even if there could be a biologically potential explanation for a real or even causal relationship between periop aspirin use and stroke (for example, by increasing periop bleeding which could increase stroke), it would NOT be an observational prognostic study to inform this hypothesis. This is not the objective of this study, and it is not an objective of prediction/prognostic studies in general, which typically do not include interventions administered during the time the cohort is followed (which are different from baseline medications) among predictors. 

I remain firm in my suggestion that perioperative aspirin should not be part of this model.

If authors wish not to take this suggestion, I recommend that they justify their choice and address this major limitation in the discussion.

[LINK]

General journal requests:

---

## [Decision Letter · Decision Letter 3]

5 Jul 2024

Dear Dr Ma,

Many thanks for submitting your manuscript "Risk Factor Analysis and Creation of an Externally-Validated Prediction Model for Perioperative Stroke Following Non-Cardiac Surgery" (PMEDICINE-D-23-03773R3) to PLOS Medicine. The paper has been reviewed by subject experts and a statistician; their comments are included below and can also be accessed here: [LINK]

As you will see, the reviewer is still concerned about the inclusion of perioperative aspirin as a risk factor in the model, which the editorial team agrees with. Therefore, after discussing the paper with the editorial team and in line with Reviewer #4's comments, we ask you to include a sensitivity analysis excluding perioperative aspirin from the model and comparing the results with the full model. We strongly feel that it is not sufficient to address the risk factor "perioperative aspirin" as a major limitation in the discussion. Please note that this is a requirement for further consideration of your manuscript at PLOS Medicine.

We ask that you submit your revision by Jul 26 2024. However, if this deadline is not feasible, please contact me by email, and we can discuss a suitable alternative.

Don't hesitate to contact me directly with any questions (atosun@plos.org). 

Best regards, 

Alex

Alexandra Tosun, PhD 

Associate Editor

PLOS Medicine

atosun@plos.org

Comments from the reviewers: 

Reviewer #4: The authors seem to have appreciated the reasoning around the very important limitations of including perioperative aspirin in their model. However, given the choice, they preferred not to rerun their study without perioperative aspirin in the model, and to discuss the inclusion of perioperative aspirin among the limitations of their study. Their choice is definitely disappointing since the inclusion of aspirin increases (I would say, not surprisingly) the AUC of their model, but dramatically reduces the ability to interpret what their model is for, and its utility and application in clinical practice when we are called to assess the risk of perioperative stroke of patients undergoing noncardiac surgery. This is a large database, to have a properly done model would have been great. "Considering the fact that this variable is highly recognizable in clinical practice" is not a sound justification to include it as a predictive factor, neither methodologically (as indicated in my previous comments) not practically, since, if the variable in the model stands for use of aspirin before and after surgery, the model can be applied only after surgery - and when should we calculate this predictor for our patient? How many days after surgery should we wait to decide whether this is a "yes aspirin" or a "no aspirin"? would a stroke have already occurred at that point?. This is to say that, if the model stays as it is, with perioperative aspirin included, authors should mitigate any claim in their paper for using the model to improve clinicians' ability to predict the risk of perioperative stroke. Theirs is a multivariable analysis for perioperative factors associated with the occurrence of perioperative stroke (including possibly after-the-fact variables). 

I would also suggest two additional revisions around the same issue:

- in the Methods, I would further clarify what is perioperative use of aspirin. Currently they say "the situation of aspirin medication both pre-operatively and post-operatively". Was this a binary variable? If so, what patients did the "no" category include? Did it include both those who were not on aspirin at baseline, and those who were on aspirin at baseline but did not continue it after surgery?

- in the Discussion of this as a study limitation, the authors added "Thirdly, due to the deficiency of retrospective database, the precise temporal relationship between perioperative aspirin medication and the occurrence of perioperative stroke was difficult to identify, which may lead to a biased analysis owing to the potential reverse association between this variable and the outcome. In addition, as an intervention administered during the perioperative period, aspirin medication could be a surrogate for high stroke risk, and the reason why aspirin was continued perioperatively should be taken into consideration during clinical application of this prediction model as it would not change clinical medication decision in these high risk patients." I suggest to rephrase it as "Thirdly, we found that perioperative aspirin was strongly associated with perioperative cover stroke. This could have been due to aspirin being a marker of a high cardiovascular risk that other variables included in the model did not fully capture; however, this could also be due reverse association. In fact, due to the retrospective nature of our analysis, based on an administrative database, we could not determine the precise temporal relationship between perioperative aspirin use and the occurrence of perioperative stroke. This represents a major limitation to the application of our model for the prediction of the risk of perioperative stroke in clinical routine."

---

* Please upload any figures associated with your paper as individual TIF or EPS files with 300dpi resolution at resubmission; please read our figure guidelines for more information on our requirements: http://journals.plos.org/plosmedicine/s/figures. While revising your submission, please upload your figure files to the PACE digital diagnostic tool, https://pacev2.apexcovantage.com/. PACE helps ensure that figures meet PLOS requirements. To use PACE, you must first register as a user. Then, login and navigate to the UPLOAD tab, where you will find detailed instructions on how to use the tool. If you encounter any issues or have any questions when using PACE, please email us at PLOSMedicine@plos.org.

---

## [Decision Letter · Decision Letter 4]

16 Dec 2024

Dear Dr. Ma,

Thank you very much for re-submitting your manuscript "Risk Factor Analysis and Creation of an Externally-Validated Prediction Model for Perioperative Stroke Following Non-Cardiac Surgery" (PMEDICINE-D-23-03773R4) for review by PLOS Medicine.

Thank you for your detailed response to the editors' and reviewers' comments. I have discussed the paper with my colleagues and the academic editor, and it has also been seen again by two of the original reviewers. The changes made to the paper were mostly satisfactory to the reviewers. As such, we intend to accept the paper for publication, pending your attention to the reviewers' and editors' comments below in a further revision. When submitting your revised paper, please once again include a detailed point-by-point response to the editorial comments.

[LINK]

In revising the manuscript for further consideration here, please ensure you address the specific points made by each reviewer and the editors. In your rebuttal letter you should indicate your response to the reviewers' and editors' comments and the changes you have made in the manuscript. Please submit a clean version of the paper as the main article file. A version with changes marked must also be uploaded as a marked up manuscript file. Please also check the guidelines for revised papers at http://journals.plos.org/plosmedicine/s/revising-your-manuscript for any that apply to your paper.

We ask that you submit your revision within 1 week (Dec 23 2024). However, if this deadline is not feasible (including due to the upcoming holidays), please contact me by email and we can discuss a suitable alternative. Please note that the editorial team will be out of office from 23 December 2024 up to and including 3 January 2025.

Please do not hesitate to contact me directly with any questions (atosun@plos.org). If you reply directly to this message, please be sure to 'Reply All' so your message comes directly to my inbox.

We look forward to receiving the revised manuscript.   

Sincerely,

Alexandra Tosun, PhD

Associate Editor 

PLOS Medicine

plosmedicine.org

Comments from Reviewers:

Reviewer #2: We thank the authors for addressing our previous comments.

Reviewer #4: My thanks and congratulations to the authors for the efforts they made to change their analysis and address the reviewer's and editors' concerns. Their work is now more solid and has a more straightforward interpretation and potential for use in clinical practice. 

A few minor suggestions:

1) The authors can now remove the definition of perioperative aspirin and its inclusion (table 1) among the patient characteristics since this is no longer a variable of relevance in their paper. 

2) The authors should clarify the statement regarding the DCA: "In the current model, the NB was 0.002 if all the subjects were treated, and the NB was 0 if none of them were treated." "Treated" with what? Do they mean "if the model was used"?

3) When they decided to translate the model into categories of risk, why dod they decide to have only one threshold and only two categories, i.e., low and high risk? To have at least 3 categories may be more informative. In any case, I believe that the average absolute risk estimate for each category is what has most clinical relevance. For this reason, in the two examples that they include in the model application section they should provide the corresponding estimated absolute risk rather than saying that their score positioned them into the low vs high risk category.

4) Page 16, lines 333-334: "which was similar to two recent comprehensive studies" should read as "which was similar to the incidence found in two recent comprehensive studies".

[LINK]

Academic Editor Comments:

On the aspirin question - the authors have taken it out of the model but still include it in their list of data collected. This is entirely appropriate, but the authors need to either:

1. Explain in the methods why they subsequently did not use the aspirin data in their model (e.g. by paraphrasing reviewer 4's discussion), or

2. Provide the aspirin results (in a supplementary section if necessary) and then explain in the discussion why they did not use them (as above).

Requests from Editors:

DATA AVAILABILITY 

Please note that the data availability statement in the online submission form and in the main text on lines 418-419 do not match. Please revise according to the following:

CODE AVAILABILITY

We expect all researchers with submissions to PLOS in which author-generated codeunderpins the findings in the manuscript to make all author-generated code available without restrictions upon publication of the work. In cases where code is central to the manuscript, we may require the code to be made available as a condition of publication. Authors are responsible for ensuring that the code is reusable and well documented. Please make any custom code available, either as part of your data deposition or as a supplementary file. Please add a sentence to your data availability statement regarding any code used in the study, e.g. "The code used in the analysis is available from Github [URL] and archived in Zenodo [DOI link]" Please review our guidelines at https://journals.plos.org/plosmedicine/s/materials-software-and-code-sharing and ensure that your code is shared in a way that follows best practice and facilitates reproducibility and reuse. Because Github depositions can be readily changed or deleted, we encourage you to make a permanent DOI'd copy (e.g. in Zenodo) and provide the URL.

TITLE

Please revise your title according to PLOS Medicine's style. Your title must be nondeclarative and not a question. It should begin with main concept if possible. "Effect of" should be used only if causality can be inferred, i.e., for an RCT. Please place the study design ("A randomized controlled trial," "A retrospective study," "A modelling study," etc.) in the subtitle (ie, after a colon).

ABSTRACT

1) l.7: We suggest clarifying that the model was built to predict the incidence of perioperative stroke after non-cardiac surgery.

2) l.11ff: PLOS Medicine prefers the use of patient-centered language, e.g. patients undergoing non-cardiac surgery (or similar). Please revise throughout the manuscript, including tables and figures (including those in the Supporting Information).

3) l.13: We suggest briefly mentioning the method used to narrow the risk factors down to the thirteen included. For example: After multivariate analysis and stepwise elimination (P<0.05), thirteen variables…” (or similar).

4) l.14: Please define ‘ASA’.

5) l.19ff: Please report statistical information as follows to improve clarity for the reader "22% (95% CI [13%,28%]; p</=). For example: “0.893 (95% confidence interval (CI) [0.879,0.908]; P<0.001)”. When reporting 95% Cis, please separate upper and lower bounds with commas instead of hyphens as the latter can be confused with reporting of negative values. Please revise throughout the manuscript.

6) ll.31-35: We suggest splitting the sentence into two sentences: In this work, we identified thirteen independent risk factors for perioperative stroke and constructed an effective prediction model with well-supported external validation in Chinese patients undergoing surgery. The model may provide potential intervention targets and help to screen high-risk patients for perioperative stroke prevention.

7) Please ensure that all numbers presented in the abstract are present and identical to numbers presented in the main manuscript text.

AUTHOR SUMMARY

1) ll.50-54: Please define all abbreviations used for the first time (ASA, MAP, FAR, FPG).

2) ll.61-62: Please revise and clarify what you mean by providing potential intervention targets.

INTRODUCTION

If there has been a systematic review of the evidence related to your study (or you have conducted one), please refer to and reference that review and indicate whether it supports the need for your study.

METHODS AND RESULTS

1) l.153: Please define ‘ICD’ at first use.

2) l.224: Please replace ‘exhibited’ with ‘experienced’.

3) ll.251-252: “As analyses were adjusted in the final model…” – adjusted for? Please specify and ensure that these adjustments are described in the Methods section.

4) ll.285-286, please revise: “The model exhibited a well-calibrated performance in the cohorts from Nanfang Hospital (Figure 2C) and Henan Provincial People's Hospital (Figure 2D),…”

5) Figure 1/2: Please provide definitions of numerical values (in parentheses). Please define 'Pr'.

6) Table 1: Please remove the p-values from Table 1. Please define ‘yr’ or write ‘years’.

DISCUSSION

General guidance: Please present and organize the Discussion as follows: a short, clear summary of the article's findings; what the study adds to existing research and where and why the results may differ from previous research; strengths and limitations of the study; implications and next steps for research, clinical practice, and/or public policy; one-paragraph conclusion.

1) l.326/341/343/345/347/387: Please temper claims of primacy of results by stating, "to our knowledge" or something similar. 

2) l.355/357: Please provide references.

3) Please remove any subheadings. The conclusion should be a continuous part of the discussion.

SUPPLEMENTARY MATERIAL

In the published article, supporting information files are accessed only through a hyperlink attached to the captions. For this reason, you must list captions at the end of your manuscript file. You may include a caption within the supporting information file itself, as long as that caption is also provided in the manuscript file. Do not submit a separate caption file.

When SI files are contained with a single file:

Please label the file as ‘S1 Supporting Information’.

Please apply alphabetical labelling to each table and figure contained within the S1 file. For example, ‘Fig A’ to ‘Fig Z’ and ‘Table A’ to ‘Table Z’.

Plain text does not need to be labelled and can just be given a title as necessary. For example, ‘Statistical Analysis Plan’.

Please cite tables/figures as ‘Fig A in S1 Supporting Information’ and/or ‘Table A in S1 Supporting Information’, for example.

Please cite plain text as, ‘Statistical Analysis Plan in S1 Supporting Information’, for example.

When SI files are uploaded as separate files:

Please label tables as ‘S1 Table’ (so on) and figures as ‘S1 Fig’ (and so on).

Any additional documents (protocols/analysis plans etc.) can be labelled as ‘S1 Protocol’, for example. Please cite items as exactly as labelled.

General Editorial Requests

---

## [Editor Report · Decision Letter 5]

22 Jan 2025

Dear Dr Ma, 

On behalf of my colleagues and the Academic Editor, David Menon, I am pleased to inform you that we have agreed to publish your manuscript "Risk Factor Analysis and Creation of an Externally-Validated Prediction Model for Perioperative Stroke Following Non-Cardiac Surgery" (PMEDICINE-D-23-03773R5) in PLOS Medicine.

I appreciate your thorough responses to the reviewers' and editors' comments throughout the editorial process. We look forward to publishing your manuscript. Editorially, there are a few remaining minor stylistic points that should be carefully addressed prior to publication. We will carefully check whether the changes have been made. If you have any questions or concerns regarding these final requests, please feel free to contact me at atosun@plos.org.

Please see below the minor points that we request you respond to:

1) Please ensure that the Data Availability Statement is presented as communicated by email. We have updated the statement in the online submission form (metadata), but please also check that your manuscript contains the correct version. The statement should be as follows: The data used and analyzed during the current study are not freely available for the ethical restriction, because the data contain potentially identifying and sensitive patient information. But the data are available from Department of Medical Service, The First Medical Center of Chinese PLA General Hospital upon reasonable request (Email: 15776734388@163.com and Tel: 086 010 66938152).

2) Please note that you have not fully revised the manuscript for use of patient-centered language (e.g. ‘perioperative stroke patients’). For example in the Abstract, ll.11-12, the sentence should be changed to: “In our hospital cohorts, 223,415 patients undergoing non-cardiac surgery were included in this study, with 525 (0.23%) patients experiencing a perioperative stroke”. Please carefully revise throughout the manuscript.

3) Abstract Conclusion: Please note that while you have stated in your rebuttal that you have split the sentences as suggested, the change was not made in the text. Please revise: “In this work, we identified thirteen independent risk factors for perioperative stroke and constructed an effective prediction model with well-supported external validation in Chinese patients undergoing surgery. The model may provide potential intervention targets and help to screen high-risk patients for perioperative stroke prevention.”

4) Please note that there are still instances of primacy claims in the text. Please temper claims of primacy of results by stating "to our knowledge" or something similar. We have highlighted two sentences below, but please check the full text:

a) ll.390-391: “This is the first large-scale patient database targeting perioperative stroke in China.”

b) ll.335-336: “It is worth noting that this is the first multicenter study investigating the incidence of perioperative ischemic stroke in patients undergoing non-cardiac surgery…”

5) Please change the title to: Risk Factor Analysis and Creation of an Externally-Validated Prediction Model for Perioperative Stroke Following Non-Cardiac Surgery: A multi-center retrospective and modeling study

6) In Table 1, please make sure that the statistical meaning is defined for all values. For example, is it median plus interquartile range (IQR) for 'Age, years'? Is it 'n (%)' for 'Sex'? Please add the information in/below the table. When adding definitions, please note that new abbreviations should be defined below the table.

Before your manuscript can be formally accepted you will need to complete some formatting changes, which you will receive in a follow up email (including the editorial points above). Please be aware that it may take several days for you to receive this email; during this time no action is required by you. Once you have received these formatting requests, please note that your manuscript will not be scheduled for publication until you have made the required changes.

PRESS

Sincerely, 

Alexandra Tosun, PhD 

Associate Editor 

PLOS Medicine